# Exploring the Use of Avatars in the Sustainable Edu-Metaverse for an Alternative Assessment: Impact on Tolerance

Yara Ahmed Mohebeldin Zaky [1,*] and Azza Ali Gameil [2]

[1] Faculty of Specific Education, Ain Shams University, Cairo 11566, Egypt
[2] Department of Curriculum and Instruction, King Faisal University, Hofuf 31982, Saudi Arabia; aelemam@kfu.edu.sa
* Correspondence: dr.yara.a.moheb@sedu.asu.edu.eg

**Abstract:** This article investigates how avatars in the metaverse can be used for peer evaluation in a sustainable educational setting. The goal is to find the best alternative modes of assessment (one-to-many/many-to-one) to evaluate the design of a sustainable educational environment. The research also explores how learners' avatars influence the development of tolerance and respect for others within the metaverse. The study involved 36 female graduates from King Faisal University's College of Education. A quasi-experimental design with two experimental groups was employed to assess performance. Through a product quality card and applying a tolerance (IPTS) scale for data collection. The preliminary findings indicate that the avatars within the (one-to-many) group demonstrated better performance and showed an increase in tolerance values after the intervention, compared to their counterparts (many-to-one) as an alternative mode of assessment, in the development of a sustainable Edu-Metaverse environment. The research validated that avatars enhance positive attitudes and behaviors, thereby providing insights for developing more effective educational interventions and contributing to enhancing the user experience and implementation of sustainable educational initiatives through a metaverse.

**Keywords:** Edu-Metaverse; sustainability; alternative assessment; tolerance; peer evaluation

## 1. Introduction

To promote sustainable development within communities, which seeks to achieve a harmonious blend of environmental, economic, and social factors, international standards have focused on the design and construction of eco-friendly schools [1]. However, some specific requirements and criteria must be met when creating a sustainable school. These include carefully selecting a location for the school that is far from sources of pollution, ensuring the safety of access roads, and incorporating surrounding trees [2].

The metaverse offers unparalleled opportunities to achieve sustainability in life because students do not have to travel to school and thus consume fewer fossil fuels. In addition, it helps spread ideas about sustainability as it can overcome spatial and temporal barriers, and learners can exchange ideas and best sustainable practices through Edu-Metaverse platforms [3]. Faster progress toward sustainable development requires the increased use of digital technologies and the dissemination of sustainable ideas within the metaverse [4,5]. Moreover, teachers can achieve meaningful educational outcomes and rewards when they use the metaverse system because this system can provide diverse tools that are not available in traditional, monotonous classrooms [6]. Hence, educational institutions are rapidly taking steps to use emerging technologies to sustain education, as immersive experiences enable students to acquire enhanced technological skills (critical thinking, creativity, collaboration, and communication), preparing them for digital citizenship and other educational benefits [7].

There is a growing need to modify and enhance educational curricula to align with scientific, social, economic, and cultural advancements. This includes integrating sustainable

development issues and concepts [1]. Adapting and updating the curricula provides an opportunity to impart a wide range of skills to students, enabling them to comprehend the principles of sustainability and its application across different domains [2]. This empowers students to learn how to sustainably manage resources and strike a balance between present and future needs.

The metaverse is expected to turn imagination into reality by converging different technologies and should be considered a means of sustainable education by promoting the dissemination of sustainable concepts and ideals, liberated from the constraints of time and place [8]. It relies on the use of avatars as virtual characters that represent users in the virtual world and can collaborate to explore and implement sustainable practices.

The metaverse has numerous advantages in the field of education, and participation in the so-called Edu-Metaverse is a new educational environment that has emerged with the advancement of metaverse technology—a shared and immersive 3D virtual space that has the potential to significantly impact education. Teachers use it to enhance interactive learning experiences [6].

Edu-Metaverse can have a positive impact on student learning outcomes [4], improve educational satisfaction [9] and engagement in cooperative learning [10], and enhance the effectiveness of theoretical and practical education [11]. However, many teachers deliver instruction within virtual worlds and evaluate their students outside of those worlds using traditional modes [6]. Traditional assessment modes have failed to implement learning assessments during the transition to remote teaching [12]. Therefore, there is a need to explore alternative modes of assessment that are suitable for these new environments and align with their immersive nature.

The purpose of using alternative modes of assessment is to measure skills that cannot be assessed through traditional evaluation modes [13]. The collaboration between avatars in the metaverse may provide a conducive environment for effectively implementing alternative modes of assessment, such as performance-based assessment modes, that both enhance students' achievements and give them opportunities to demonstrate their understanding and use multiple skills [14]. Evaluating the quality of a product design involves comparing a student's performance or output to certain criteria via a rubric. When students collaborate and exchange ideas, they can showcase their learning through presentations, projects, or discussions [15]. Researchers have indicated the need to design alternative modes of assessment and integrate them into remote virtual training programs [12]. Moreover, they have recommended the integration of assessment tools within virtual worlds and suggested that students' avatars should be assigned complex tasks within the virtual environment [16].

With the increasing diversity of communities in the metaverse, which is characterized as a multicultural and racially diverse world [3], there is a need to study tolerance among avatars in the Edu-Metaverse environment. In this study, the researchers highlight that tolerance is the acceptance and respect of diverse beliefs, practices, and differences without necessarily agreeing with them, which is crucial for fostering understanding and harmony in different societies. Moreover, it is an ethical value, as mentioned by Hjerm et al. [17], that should be nurtured and applied in our lives. It involves behaving well, respecting the differences of others, and, as Cuadrado et al. [18] explain, being tolerant of individuals who hold ideas and values that do not align with our own. This diversity and the practice of tolerance can enrich the user experience in a sustainable Edu-Metaverse by creating immersive and interactive educational experiences.

There are two research questions:

Q1: What is the impact of avatar characters as an alternative assessment mode (many-to-one pairing/one-to-many pairing) used in the sustainable Edu-Metaverse on product quality among graduate students?

Q2: What is the impact of avatar characters as an alternative assessment method (many-to-one pairing/one-to-many pairing) used in the sustainable Edu-Metaverse on the tolerance and respect for others among graduate students?

**Hypothesis (H1).** *There will be no statistically significant differences ($p \leq 0.05$) in the average scores on the product evaluation card of the two experimental groups due to using avatar characters as an alternative assessment mode (one-to-many pairing/many-to-one pairing) in the sustainable Edu-Metaverse.*

**Hypothesis (H2).** *There will be no statistically significant differences ($p \leq 0.05$) in the average scores of tolerances and respect for others among the two experimental groups due to using avatar characters as an alternative assessment method (one-to-many pairing/many-to-one pairing) in the sustainable Edu-Metaverse.*

Therefore, the current research aims to explore the use of student avatars in the Edu-Metaverse as a peer-based alternative assessment tool that can be utilized to promote tolerance and respect for others. Avatars participate in assessing others based on their performance, without bias based on their real-life identity, in the context of higher education. As students progress through higher academic levels, they are expected not only to acquire knowledge but also to develop interpersonal skills such as tolerance and respect for diverse perspectives.

There are several parts discussed. Section 2 presents the history and background of the use of avatars in the Sustainable Edu-Metaverse for alternative assessments and their impact on tolerance. Section 3 explains the methodology and the data validation method. Section 4 is the discussion. Finally, we conclude our work in Section 5.

## 2. Literature Review

The use of avatars in the metaverse as an alternative mode of assessment in education and the impact of their use on the development of values, tolerance, and respect is an emerging field of research. Although the literature in this specific area is limited, some relevant studies shed light on related topics and provide valuable insights.

### 2.1. Avatars in the Sustainable Edu-Metaverse

Metaverse and advanced technologies can drive sustainable education by fostering innovative learning and promoting sustainable values [19]. Sustainable education relies on the social and emotional engagement of students, enabling them to achieve better learning outcomes in virtual, face-to-face, and online environments [11]. This principle applies to learning from the metaverse, which can bring about a significant transformation in traditional teaching and learning modes [8]. Thus, supporting positive interactions in the metaverse is essential for a balanced ecosystem [6]. The Edu-Metaverse allows for training and skill development, reducing the risks of training [19]. Moreover, good design helps bridge the digital divide and achieve justice, equality, access, and sustainability [3]. A cooperative approach can inspire innovative solutions, promote sustainability, and allow learners to experience the outcomes of their learning and decisions through interactive simulations and collaborative projects [11]. Additionally, educational institutions can harness the potential of the metaverse to integrate sustainable practices into their curricula [10]. By doing so, we can develop academically distinguished learners who actively participate in building a more sustainable future [7]. For example, students can explore renewable energy systems, simulate the impact of deforestation, or even join in global climate conferences [2], all from the comfort of their homes.

The metaverse is one of the most exciting technological advancements, enabling new and enhanced ways of living, entertainment, work, and education [20]. Nagendran et al. [21] emphasize that the metaverse allows for the creation of effective experimental learning environments that foster both academic and personal development. Inceoglu and Ciloglugil [22] further explain the importance of using the metaverse in education, as it enables realistic learning experiences, enhances remote learning effectiveness, and improves student interactions with teachers and educational materials. Chanda et al. [23] consider the metaverse to be among the key technological platforms that should be integrated into learning sys-

tems. According to Chafiq et al. [24], the rapid spread of remote learning has led to the proliferation of virtual environments for educational purposes, such as the Edu-Metaverse.

Edu-Metaverse is explained by Jang and Kim [11] as a collection of internet-connected virtual worlds that contribute to creating purpose due to their multi-user social nature. Wang and Shin [19] define the Edu-Metaverse that has emerged with the advancement of metaverse technology—a shared and immersive (3D) virtual space that has the potential to significantly impact education. Provides learners with a higher degree of creativity and freedom than many other teaching modes [19]. It provides access to a wide range of resources, including multimedia presentations and interactive objects that support lesson delivery, videos, images, and audio recordings [23]. Zhang et al. [8] mention that the use of the metaverse in education facilitates collaboration between students and teachers and also provides comprehensive assessment modes [19].

Avatars play a pivotal role in the metaverse, enhancing learner engagement. Zimmermann et al. [25] explained avatars as digital representations of users' personal, physical, and demographic characteristics, which may also reflect social norms. These images have flexible attributes that can be modified, such as facial features, expressions, body shapes, skin colors, and clothing [26]. Avatars help users understand and exchange social signals and facilitate communication and interaction, including in educational activities, thus enhancing learning experiences [10]. J. Thomason and E. Ivwurie [20] contend that avatars possess similar ethical rights and obligations to that of their real counterparts. Research substantiates their benefits in educational contexts. A study by Jang and Kim [11] explored the impact of learners using avatars in metaverse environments. The findings highlighted the significance of metaverse-based education as an alternative to overcome traditional learning limitations and emphasized the importance of learners utilizing avatars.

In a study conducted by Grivokostopoulou et al. [27], the impact of students' perceptions of avatars in a virtual learning environment was investigated. Results indicated that students' participation as avatars led to higher achievement. In a study by Li et al. [10], the impact of metaverse platforms on collaborative learning task performance in an educational setting was examined. The results showed that learners' use of avatars in the metaverse increased communication, collaboration, and a sense of presence. Nagendran et al. [21] investigated the impact of avatar use on learner behavior. Results revealed that learners using avatars interacted naturally and exhibited behavioral characteristics similar to those they would display in the real world. J. Thomason and E. Ivwurie [20] suggested designing avatars specifically for educational purposes within virtual environments. A study by Zhu and Yi [28] examined user behaviors in educational metaverse applications. The results indicated that high similarity between users' avatars leads to increased task engagement and better performance.

Previous research has established the importance of avatars in enhancing learning experiences within virtual environments. The study aims to explore the potential of using learner avatars as a peer assessment tool in the Edu-Metaverse. By doing so, it seeks to develop an alternative assessment method and investigate its impact on fostering tolerance and respect among learners.

## 2.2. Alternative Modes of Assessment in the Edu-Metaverse

In a rapidly changing and evolving world, educational institutions and organizations face diverse challenges that require innovation and continuous development. One of these challenges is assessing learner performance to ensure the attainment of desired goals and sustainable excellence. According to Ahmad et al. [29], education is a lifelong, continuous process aimed at bringing about positive changes in society and individual positions. Gallardo [30] states that the assessment of the educational process provides an accurate means to examine progress and understand students' success in achieving educational objectives. As a result, assessment practices must be improved, as they have a significant impact on the quality of the learning process [14]. However, traditional assessment modes are insufficient for measuring learning and skills unless they are integrated with performance-

based assessments, which offer a more accurate evaluation of achievements [12]. This is where alternative assessments come into play [13]. The main purpose of employing alternative assessment modes is to measure skills that cannot be assessed through traditional evaluation modes [29,31]. Alternative assessments can be categorized into five categories: self-review, observation-based assessment, communication-based assessment, paper-and-pencil assessment, and performance-based assessment.

The current research focuses on performance-based assessment, defined by Braun [32] as a student's ability to articulate their work by employing their skills in real-life situations. Villarta et al. [33] state that performance-based assessment is an evaluative activity aimed at revealing students' understanding of concepts, skills, and ideas. The purpose of alternative assessments using performance-based modes is to provide an opportunity for students to demonstrate their knowledge and comprehension of a wide range of information [31]. Suastra and Menggo [14] clarify that performance assessment consists of a set of tasks assigned to learners in which a standardized assessment mode l evaluates learners' performance or production through aspects including indicators, weights, grades, and descriptors for each level of teacher preparation [13]. The reason alternative modes of assessment are used in the classroom is that they uncover what students can do rather than just what they do [15]. Therefore, as VanTassel-Baska [34] suggests, performance assessment requires test designers and other evaluators to be creative.

Virtual environments provide an immersive platform that allows for the design of interactive assessments capable of effectively measuring students' knowledge and skills. According to Al-Abdullatif [15], many teachers employ learning activities within virtual worlds but use traditional modes to assess their students outside the virtual realm. In such cases, assessment is tied to reflections on events within the virtual world rather than real-time assessments during students' performance of tasks within the virtual environment. Traditional assessment modes have failed to implement learning assessments during the transition to remote teaching [12]. The integration of virtual environments into education presents an exciting opportunity to revolutionize the assessment process and enhance students' comprehensive learning experiences.

Virtual environments rely on the use of avatars to represent learners, teachers, or other characters [25]. Collaboration between avatars in virtual environments, representing peer assessment, may provide a conducive environment for the effective implementation of alternative assessments, which is the goal of the current research. When students collaborate and exchange ideas, they can demonstrate their learning through projects or discussions, enabling a more comprehensive assessment of student knowledge [16], that is, one that considers not only the product but also the process of working together and developing important social skills [35]. Reviewers can be peers within the group or learners from other groups or disciplines [36]. Cooperative learning also enhances the effectiveness of alternative assessments by promoting critical thinking, problem-solving, and communication skills [37]. These immersive and interactive practices allow for real-time assessment and feedback, enabling students to present their understanding and skills in a more realistic and meaningful way [38]. Students who engage in design and assessment can improve their learning outcomes.

The metaverse is one of the evolving 3D virtual environments that dominate the education scene [39]. It provides a unique digital space that enables students to engage in immersive learning experiences that go beyond the boundaries of traditional assessment and in which learner avatars collaborate [16]. In such environments, students can participate in practical simulations, role-playing, and virtual experiments, and their performance can be assessed based on their decision-making abilities, critical thinking skills, and problem-solving competencies. Hence, students actively contribute to their learning processes and are evaluated based on their collaborative efforts.

The research topic has been extensively explored in prior studies, which have examined whether avatars as part of alternative modes of assessment are significant for developing learners' skills [35]. According to Bosch and Ellis [40], the continuous use

of avatars is an important tool for teacher development. Avatar experiences not only enhance task and behavioral management but also foster the soft skills necessary for a sustainable and successful teaching career [11]. This is available in the Edu-Metaverse environment. Avatars can play a role in peer assessment as part of alternative modes of assessment [30]. Individuals can receive evaluations from their virtual peers who work closely with them and interact with them, which improves the accuracy and comprehensiveness of the performance evaluation process [38]. Peer assessment activities focus on the students, encouraging them to interact with each other and consider peer comments to understand their areas of improvement. These modes can lead to better educational outcomes through peer assessments and organized feedback [35,41].

Peer assessment has been proven effective in enhancing student performance through various investigations. Crisp [16] suggested the need to develop avatars in 3D virtual learning environments by integrating assessment tools and assigning complex tasks to student avatars within the virtual world. Chang et al. [38] agree that peer assessment has an educational efficacy on student performance. The students also exhibited higher self-efficacy and critical thinking tendencies than those using traditional teacher feedback in the virtual reality design system. Yuan and Gao [42] focused on using avatars to facilitate peer learning in a virtual reality classroom, and the results showed that peer avatars helped learners interact in real time.

Bosch and Ellis [40] state that there was a positive relationship between avatar intervention and teachers' self-efficacy in classroom management, teaching strategies, and student engagement, and the study recommended further research on experimental designs of avatars. Pál and Koris [12] stated the need to design alternative modes of assessment and incorporate them into remote training programs to accommodate the changing educational environment. Doing so requires a review of assessment practices and curricula to identify best practices for alternative assessment modes. Figure 1 depicts alternative modes of peer assessment in the Edu-metaverse using avatars (one-to-many/many-to-one) as presented in the research.

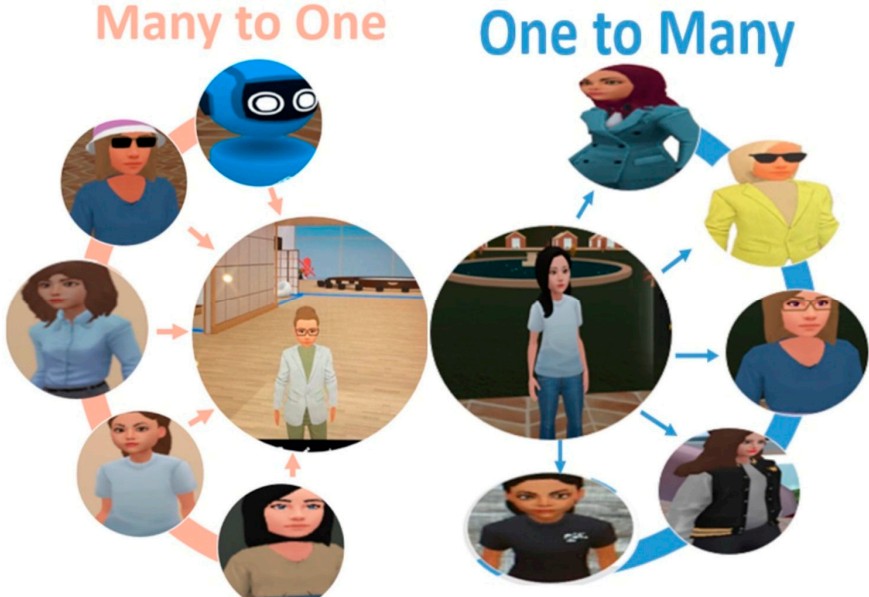

**Figure 1.** Alternative modes of assessment peers (one-to-many/many-to-one).

Therefore, this study will delve deeply into the use of avatars to represent students in the Edu-Metaverse as a tool for determining whether a group of avatars collaborating to evaluate a single avatar or a single avatar evaluating a group of peers is a more appropriate alternative mode of assessment. Based on an assessment rubric prepared by the teacher.

*2.3. Tolerance and Respect for Others*

With the increasing diversity of communities, accepting and respecting others have become important elements in ensuring individual well-being and societal development. In the world of the metaverse, avatars represent individuals and their interactions in the virtual world, in which many people from different races, cultures, and beliefs participate. Research has emphasized the importance of promoting positive values in the metaverse, such as caring for the well-being of others, respecting independence, and fostering tolerance among students [20]. The avatars present in the metaverse can create immersive and interactive environments that enhance the exploration and understanding of these values. By engaging in virtual worlds, students can take on different roles, leading to an understanding of diverse perspectives [28]. Sakalli et al. [43] argue that, with these developments, students are becoming digital citizens who master technology in the educational process.

Tolerance and respect are closely linked to the development of values and can be influenced by educational experiences. Sneka and Ramasamy [44] define tolerance as the ability to accept, understand, acknowledge, respect, and appreciate the social, political, and religious perspectives of another person. The personal values of an individual are shaped through communication, interaction with cultures, and their integration into a complete social space. Hjerm et al. [17] consider tolerance to be a human act that we must nurture and apply in our daily lives. In the authors' view, tolerance involves acting well and respecting differences to embrace the diversity that makes us stronger and the values that unite us. Sakalli et al. [43] point out that tolerance training is necessary for individuals to recognize their boundaries, acknowledge the need for human existence in social life, and direct their intelligence toward goodness. Budnyk et al. [45] explain that tolerance is an attitude toward culture, physical differences among people, or diverse opinions based on tolerating those who are "not like the others". Cuadrado et al. [18] discussed UNESCO's principles of tolerance, published in 1995, emphasizing the importance of accepting diversity and respecting and appreciating others. Tolerance in society is built upon four distinct concepts, namely, permission, coexistence, respect, and appreciation, and involves being tolerant of individuals who have ideas and values that do not align with one's own.

The need to teach tolerance to learners was highlighted by events following the COVID-19 pandemic; there has been a mandatory shift to remote learning in schools worldwide. Participation in online classes has been significantly high due to the virtual adaptation of technology in educational institutions and students' enjoyment of the experiences. Sakalli et al. [43] highlight the potential impact on students' moral values when teachers introduce them to out-of-school learning environments, such as virtual environments. Grivokostopoulou et al. [27] point out that virtual reality environments, including the metaverse, which encompasses avatars, can enhance communication among student groups and reduce bias. Avatars in the metaverse provide opportunities for students to interact with individuals from different backgrounds and cultures, promoting tolerance and respect through increased exposure and understanding [46].

The use of the metaverse in education aligns with learner-centered principles. It plays an active role in learners' collaborative and technology-integrated education and enables students to engage in critical thinking, creativity, and tolerance simultaneously [20].

The existing literature has provided valuable insights into the research topic, with studies focusing on the importance of fostering tolerance in the educational process. Sneka and Ramasamy [44] discussed how using technology in teaching tolerance increases students' attention and engagement during lessons, focuses on sustainable learning, and stimulates understanding of the concept of tolerance through a cooperative and enjoyable learning environment. Budnyk et al. [45] agree that cooperative work can be achieved when students interact with each other, teachers, and content with a spirit of tolerance, openness, empathy, and satisfaction in the workplace, leading to the success of the educational system. Lim and Lee [46] examine the tolerance of individuals with special educational needs. Their results highlighted that tolerance requires respect for other groups despite their differences.

The values of tolerance in peer education include mutual assistance, empathy, cooperation, and respect for diversity. Peer education modes emphasize the importance of instilling tolerant attitudes through collaboration and social partnerships [47]. The objective of tolerance education, rooted in the value of respecting others, is to encourage openness while cultivating reciprocity and fostering sustainable engagement amidst diverse perspectives [48]. Education for tolerance has become increasingly vital in the diverse societies of today, and the effectiveness of tolerance education is enhanced by integrating local values and cultural contexts to improve social communication and empathy [49]. Ultimately, peer education should focus on promoting respect, mutual recognition, and understanding in various virtual environments, including the metaverse.

Studies have shed light on the influence of peers in virtual environments on promoting tolerance. Liebkind and McAlister [48] argue that tolerance levels increased in peer-interacting groups. Windasari and Dimyati [47] agree that peer attachment and tolerance have an impact on the social competence of teenagers and can also be strong indicators of social competence. Tanyel and Kıralp [50] state that peer tolerance training programs were effective in increasing tolerance inclinations and human values among students.

From the above, it is evident that tolerance is a state of respecting and appreciating differences, religious beliefs, and differing opinions, establishing good relationships with people, and being able to act justly without bias [43]. In this context, accepting, respecting, and valuing diversity are important aspects of achieving a positive, comprehensive, and sustainable experience in the metaverse. There should be tolerance and respect for individuals as well as their cultures and beliefs, and there should be no prejudice or discrimination. Therefore, the current study aims to use avatars in the Edu-Metaverse as alternative assessment tools and explore their impact on the extent of tolerance and respect for others, as shown by students' avatars.

## 3. Materials and Methods

### 3.1. Methodology

Because this study explores cause-and-effect relationships in the metaverse as a virtual world that simulates the real environment [41], it employed a quasi-experimental research methodology. This study explores the influence of avatars as an alternative assessment method in the metaverse (the cause) on the development of a sustainable Edu-Metaverse environment and the promotion of tolerance and respect (the effect) among graduate female students.

The study employed a two-group experimental research design. To ensure the homogeneity of the two groups, an IPTS pre-test was applied (Table 1).

**Table 1.** Mann–Whitney U test to assess the equivalence of the two groups (one-to-many/many-to-one) on the modified IPTS before the intervention implementation.

| Group | N | Mean Rank | Sum of Ranks | Mann-Whitney U | Z | *p*-Value |
|---|---|---|---|---|---|---|
| Many-to-one (pre) | 18 | 15.39 | 277.00 | | | |
| One-to-many (pre) | 18 | 21.61 | 389.00 | 106.000 | −1.788 | 0.074 |
| Total | 36 | | | | | |

Group 1: Avatars representing female students participated in evaluating the sustainable learning environment designed by each student in a (one-to-many) assessment formats.

Group 2: Avatars representing female students participated in evaluating the sustainable learning environment designed by each student in a (many-to-one) assessment format.

### 3.2. Participants

The data used in this study were collected from a random sample of female students enrolled in the master's program in Educational Technology at King Faisal University in the

Kingdom of Saudi Arabia (*n* = 36). The students' ages ranged from 22 to 35 years, and they were taking a course entitled "Three-Dimensional Educational Graphics", Participants had no background in using metaverse. They were divided into two groups, each consisting of 18 students. This study was conducted during the first semester of the academic year 2023/2024.

### 3.3. Data Collection Instrument

### 3.3.1. Tolerance and Respect for Others Survey

The tolerance and respect for others survey was adapted from Thomae et al. [51]. The original interpersonal tolerance scale (IPTS) consisted of 34 closed-ended items, including four reversed items. The survey used a 4-point Likert scale (strongly agree = 4; agree = 3; disagree = 2; strongly disagree = 1). The range of scores in the survey was between 34 and 136. A total of five experts from the educational technology field evaluated the instrument's validity. The experts recommended rephrasing some of the items for clarification. The recommended modifications were implemented in the form of the instrument, hereafter referred to as the modified IPTS.

The reliability of the original IPTS ranged between 0.77 and 0.86. The reliability of the modified IPTS was calculated using Spearman's coefficient modes. The Spearman's coefficient showed a result of 0.92, which indicated high and significant reliability.

### 3.3.2. Product Quality Card

Evaluating the quality of a product design involves comparing a student's performance or output to certain criteria via a rubric [16]. To evaluate the performance of peer group avatars in the metaverse as an alternative mode of assessment, a product quality card was created by the teacher to assess the quality of the product design of the sustainable Edu-Metaverse environments produced by the students. The teacher's evaluation score served as the benchmark for assessing both experimental groups.

The card included four main axes (interaction and participation, educational suitability, ease of use and interface, sustainable and scalability) for a total of 20 items that were evaluated on a 4-point Likert scale (novice = 1; emerging = 2; developing = 3; proficient = 4). The scores on the evaluation product quality card ranged from 20 to 80 points, and the overall score reflected the quality of the product. The validity of the product quality card was assessed by a panel of experts in the field of educational technology who recommended rephrasing certain items for clarification. There was a good consensus on the overall validity of the card. The reliability of the product quality card was calculated from the data collected in the pilot study using the Alfakronbach coefficient, which was 0.96, indicating extremely high reliability.

### 3.4. Experimental Procedure and Data Collection

- Ethical approval was obtained from the Research Ethics Committee at King Faisal University (Reference KFU-REC-2024-JUN-ETHICS1829).
- The 36 graduate students were introduced to the concept of the Edu-Metaverse and familiarized themselves with some applications of the metaverse that can be used in education. Then, the students were randomly divided into two groups, each consisting of 18 students. The practical application was conducted using the Frame VR platform.
- As the selected metaverse platform, Frame VR, only allows free access to seven participants (avatar images) in a single environment, each group was divided into three subgroups consisting of six students. Each student had an avatar and there was also a teacher avatar. The first group of avatars represented the female students evaluating the sustainable Edu-Metaverse environments in a (one-to-many) assessment formats. The avatars of the second group, on the other hand, participated in evaluating it as a many-to-one assessment.
- The teacher conducted an introductory session with the students to show them how to use the Frame VR platform to deliver virtual lectures to enhance the learning

experience. Students were guided to choose and design avatars that would represent their personas in the metaverse. The platform's tools, such as the whiteboard, video files, PDF files, and ability to showcase different websites, were used to explain the key components of the platform and familiarize students with its design and the skills they needed to develop for successful project completion. The content delivered by the teacher was the same across all groups.

- The students were individually tasked with designing an educational environment using metaverse tools to explain the concept and types of sustainability.
- The concept of environmental sustainability, which focuses on preserving and protecting the natural environment and its resources. Such as reducing carbon emissions, conserving energy and water, promoting biodiversity, and minimizing pollution and waste.
- The concept of economic sustainability, which emphasizes the importance of maintaining a strong and resilient economy over the long term. It involves strategies that foster economic growth and create employment opportunities.
- The concept of social sustainability emphasizes the well-being and quality of life of individuals and communities. It involves promoting social justice, equality, inclusivity, and access to basic needs such as education, healthcare, and housing.
- The teacher designed a product quality card form as an alternative assessment tool to measure task performance and then trained the students on how to use it for evaluation.
- All participants were requested to upload the link to their designed sustainable Edu-Metaverse environment onto the discussion board within the teacher's metaverse environment on the Frame VR platform.
- The teacher shared the product quality card link and the link to the modified IPTS with the students through the chat feature available on the metaverse platform. Avatars in both groups were requested to post their feedback and opinions about the environment they were evaluating and how it could be improved, using the discussion tools and the modes to evaluate the design of the educational sustainable metaverse according to their respective groups. The peers posted their feedback on the discussion board.
- The students' performance evaluations in designing the product (Edu-Metaverse environments) were analyzed. To assess the quality of the environment, a product quality card was designed based on a review of the relevant literature [15]. This four-part evaluation card was used to describe the performance of avatars within the metaverse environment and comprised four main evaluation dimensions (interaction and engagement; educational alignment; ease of use and interface; and sustainable and scalability).
- The implementation of the research experiment took approximately four weeks, from 1 December 2023 to 27 December 2023.
- The product quality card and the modified IPTS were applied once the experiment was completed. Subsequently, grades were prepared using tools, and the data were collected for statistical analysis. Figures 2 and 3 show examples of sustainable Edu-Metaverse environments designed by the participants.

*3.5. Data Analysis, Results, and Discussion*

In this study, we identified the impact of using avatar characters in the sustainable Edu-Metaverse as a tool for peer assessment instead of traditional assessment modes. We employed an alternative assessment approach that used one-to-many and many-to-one peer matching based on a product quality card prepared by the teacher. We then examined how this influenced the development of tolerance, values, and respect for others among graduate students.

The modified IPTS was pre-assessed before the experimental intervention was carried out to ensure homogeneity between the two experimental groups. The Mann–Whitney U test was used to determine the significance of differences in the mean ranks of responses between the two groups (Table 1).

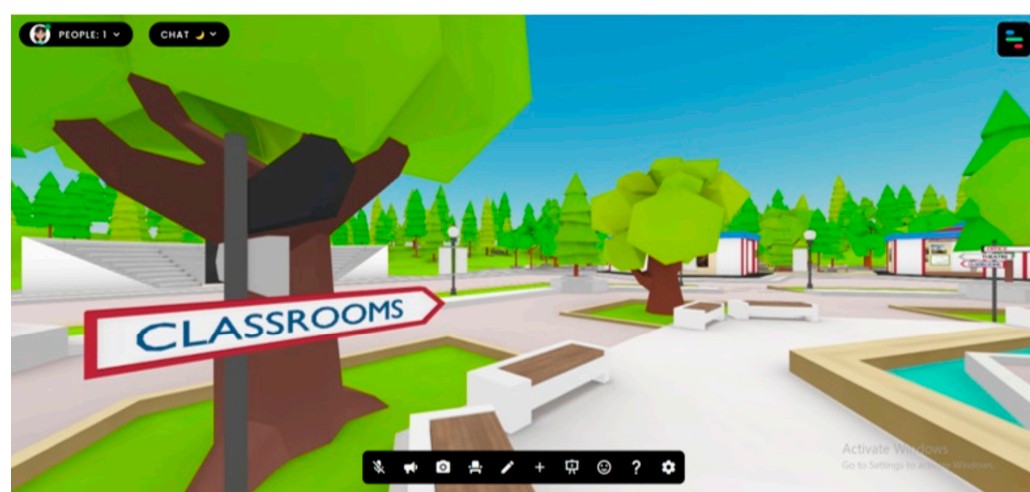

**Figure 2.** Example #1 of students' sustainable Edu-Metaverse environments (the concept of environmental sustainability).

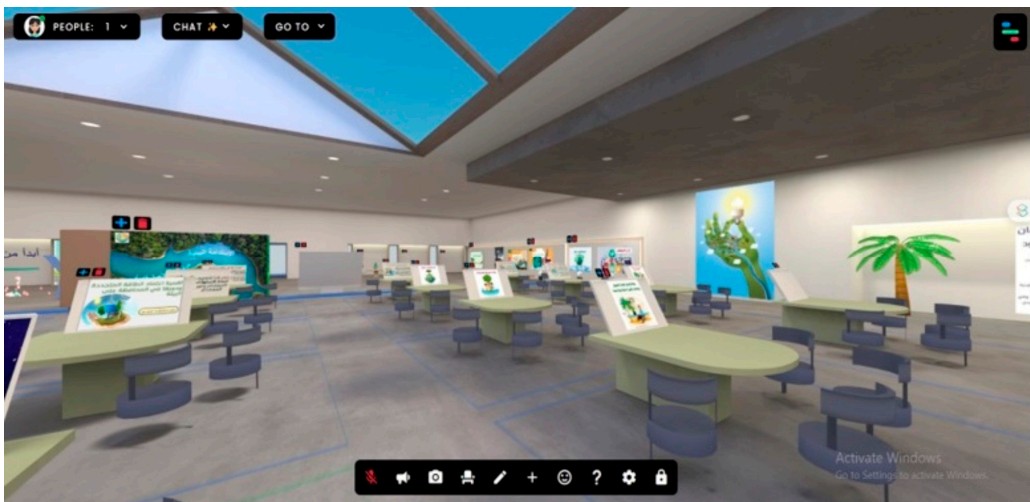

**Figure 3.** Example #2 of students' sustainable Edu-Metaverse environments (the concept of social sustainability).

As shown in Table 1, the Mann–Whitney test revealed that there are no statistically significant differences between the average ranks of the scores of the two groups in the pre-application of the modified IPTS ($p \leq 0.05$); hence, they were considered homogeneous.

Arithmetic means and standard deviations were also calculated for the evaluation scores on the product quality card according to peer evaluation (one-to-many/many-to-one) and teacher evaluation, and Table 2 shows these results.

**Table 2.** Means and standard deviations between the evaluation scores of the two experimental groups and the teacher on the product quality card.

| Teacher Evaluator | Group Evaluator | N | Mean | Std. Dev. | DIFF. |
|---|---|---|---|---|---|
| | Many-to-one | 18 | 79.11 | 1.231 | 8.28 |
| Teacher/Many-to-one | | 18 | 70.83 | 2.256 | |
| | One-to-many | 18 | 74.11 | 5.028 | 3.33 |
| Teacher/One-to-many | | 18 | 70.78 | 2.315 | |

Table 2 shows that the evaluation of the many-to-one group had an arithmetic mean of (79.11) and a standard deviation of (1.231), and the evaluation of the teacher/many-to-one group had an arithmetic mean of (70.83) and a standard deviation of (2.256), with a difference of (8.28) degrees. The evaluation of the (one-to-many) group had an arithmetic mean of (74.11) and a standard deviation of (5.028), and the evaluation of the teacher one-to-many group had an arithmetic mean of 70.78 and a standard deviation of 2.315, with a difference of 3.33 degrees. It is noted that the evaluation of the (one-to-many) group was close to the evaluation of the teacher–one-to-many group. When comparing the arithmetic mean and standard deviation between the two groups evaluations, we notice a difference of 5.00 degrees.

To demonstrate the significance of the differences between the average ranks of the evaluation scores of the two experimental groups on the product quality card between the evaluation (one-to-many/many-to-one) and the teacher evaluation, the Wilcoxon test was used, as shown in Table 3.

**Table 3.** Wilcoxon test shows the significance of the differences between the average ranks of the peer evaluation scores (one-to-many/many-to-one) and the teacher's evaluation on the product quality card.

| Evaluator | | N | Mean Rank | Sum of Ranks | Z | *p*-Value |
|---|---|---|---|---|---|---|
| Teacher/Many-to-one | Negative Ranks | 18 | 9.50 | 171.00 | −3.734 | 0.000 |
| | Positive Ranks | 0 | 0.00 | 0.00 | | |
| Teacher/One-to-many | Negative Ranks | 14 | 9.14 | 128.00 | −1.858 | 0.063 |
| | Positive Ranks | 4 | 10.75 | 43.00 | | |

Table 3 shows that there were statistically significant differences at the level of significance of 0.05 between the average ranks of the evaluation scores of the study sample members for the group Teacher/many-to-one and in favor of the peers, meaning that the evaluation of the group's peers, many-to-one, is higher than the evaluation of the teacher on the product quality card. It also shows that there were no statistically significant differences at a significance level of 0.05 between the average ranks of the peer evaluation scores in the (one-to-many) group and the teacher evaluation.

To show the significance of the differences between the average ranks of the evaluation scores of the first group (many-to-one) and the second group (one-to-many) on the product quality card, the Wilcoxon test was used, as shown in Table 4.

**Table 4.** Wilcoxon test to show the significance of the differences between the average ranks of the evaluation scores of the many-to-one and one-to-many groups on the product quality card.

| Evaluator | | N | Mean Rank | Sum of Ranks | Z | *p*-Value |
|---|---|---|---|---|---|---|
| One-to-many/many-to-one | Negative Ranks | 14 | 7.50 | 105.00 | −3.448 | 0.001 |
| | Positive Ranks | 0 | 0.00 | 0.00 | | |
| | Ties | 4 | | | | |
| | Total | 18 | | | | |

Table 4 shows that there were statistically significant differences at a significance level of 0.05 between the average ranks of the peer evaluation scores in the (one-to-many) group and the teacher's evaluation. Its evaluation was better in terms of accuracy and closeness to the teacher's evaluation, while the many-to-one group's evaluation was higher and farther away from the teacher's evaluation on the product quality card. Thus, the first hypothesis was rejected.

To demonstrate the significance of the differences between the average ranks of the two groups' scores (one-to-many/many-to-one) on the modified IPTS post-intervention, the Mann–Whitney test was used, as shown in Table 5.

**Table 5.** Mann–Whitney test to show the significance of the differences between the average ranks of the two groups' scores (one-to-many/many-to-one) on the modified IPTS administered post-intervention.

| Group | N | Mean Rank | Sum of Ranks | Mann-Whitney U | Z | *p*-Value |
|---|---|---|---|---|---|---|
| Many-to-one (post) | 18 | 16.83 | 303.00 | | | |
| One-to-many (post) | 18 | 20.17 | 363.00 | 132.000 | −0.956 | 0.339 |
| Total | 36 | | | | | |

Table 5 shows that there were no statistically significant differences at the significance level of 0.05 between the average ranks of the scores of the members of the two groups in the modified IPTS administered post-intervention.

Arithmetic means and standard deviations were also calculated for the evaluation scores on the modified IPTS according to peer evaluation in the two groups (one-to-many—many-to-one) before and after the intervention. The results are shown in Table 6.

**Table 6.** Arithmetic means and standard deviations between the evaluation scores of the two experimental groups on the modified IPTS before and after the intervention.

| Evaluator | Mean | Std. Deviation |
|---|---|---|
| Many-to-one (pre) | 90.11 | 7.984 |
| Many-to-one (post) | 98.11 | 16.496 |
| One-to-many (pre) | 86.11 | 3.596 |
| One-to-many (post) | 91.50 | 3.989 |

Table 6 shows that the assessment of the many-to-one group on the modified IPTS before the intervention had an arithmetic mean of 90.11 and a standard deviation of 7.984, while that after the application had an arithmetic mean of 98.11 and a standard deviation of 16.496. As regards the evaluation of the (one-to-many) group, the pre-intervention modified IPTS had an arithmetic mean of 86.11 and a standard deviation of 3.596, and the post-intervention had an arithmetic mean of 91.50 and a standard deviation of 3.989.

To demonstrate the significance of the differences between the average ranks of the assessment scores of the two experimental groups (one-to-many/many-to-one) on the modified IPTS between the assessments before and after the intervention, the Wilcoxon test was used. The results are shown in Table 7.

**Table 7.** Wilcoxon test to show the significance of the differences between the average ranks of the peer evaluation scores (one-to-many/many-to-one) in the modified IPTS before and after the intervention.

| Evaluator | | N | Mean Rank | Sum of Ranks | Z | *p*-Value |
|---|---|---|---|---|---|---|
| Many-to-one (post) Many-to-one (pre) | Negative Ranks | 0 | 0.00 | 0.00 | | |
| | Positive Ranks | 7 | 4.00 | 28.00 | −2.384 | 0.017 |
| | | 11 | | | | |
| One-to-many (post) One-to-many (pre) | Negative Ranks | 18 | 0.00 | 171.00 | | |
| | Positive Ranks | 0 | 9.50 | 0.00 | −3.732 | 0.000 |
| | Ties | 0 | | | | |

Table 7 shows that there were statistically significant differences at the level of significance of 0.05 between the average ranks of the group evaluation scores, many-to-one, in the pre- and post-applications on the tolerance scale, in favor of the pre-application, while the group evaluation scores showed (one-to-many), in favor of the post-intervention. Thus, the second hypothesis was rejected. The difference can be attributed to the use of avatar characters as an alternative assessment mode (one-to-many pairings/many-to-one pairings), in favor of (one-to-many) pairings.

## 4. Discussion

The results of the test of the first hypothesis reflect that the influence of avatars in the metaverse environment had an impact on the students' assessment of their peers, as scores in the one-to-many group were closer to the teacher's evaluations of the product quality card for a sustainable educational environment in the metaverse (Table 4). This may be attributed to the metaverse's educational design, with the positive impact of peer review contributing to enhanced learning outcomes, as supported by [16,35–37]. The appropriate design and integration of educational factors in the educational metaverse helps to interact with learners' avatars enjoyably, improves the learner's experience, and enhances participation, which may lead to achieving higher grades. Which may account for the observed outcomes.

Additionally, the chosen platform, Frame VR, allowed for a closed environment that included only the teacher and learners in the form of avatars. The sample consisted of female graduate students who were familiar with each other. The environment was supervised and monitored by the teacher to provide a safe educational environment for the students, which may have had an impact on the equality of tolerance values between the two groups in the intervention. However, the teacher did not intervene during the evaluation process and allowed the avatars to freely interact.

Furthermore, the pre-existing familiarity among the participants could have influenced their interaction. Prior acquaintance may lead to the formation of strong bonds between avatars. Consistent with [8,26,27], using an avatar in the metaverse helps to overcome the limitations of traditional education and increases cooperation within the Edu-Metaverse environment.

The results can also be attributed to the learners being influenced by their teacher and emulating the teacher's approach and values during their training when using the product quality card to assess their peers. Although both groups demonstrated tolerance and respect for others' values after the intervention, these were higher among the (one-to-many) peer group after its members had used the metaverse. These findings corroborate previous research [9,10,21] on the advantages of peer-as-avatar cooperation in sustainable Edu-Metaverse settings.

This result is also consistent with the Vygotskian sociocultural theory, which emphasizes that learning is an active and social process. According to this theory, learners collaborate with their peers to help them progress [35]. Consistent with the interpretation Korthagen [52] that aligns with situated learning theory, learning occurs through applying knowledge in a specific context and through interaction and participation in learning communities. Situated learning requires the provision of authentic or realistic learning environments, and the metaverse offers a unique simulation of such environments. This may be attributed to the teacher's satisfaction with students' performance during interactions within the metaverse, which led students to accept the work of others and instilled in them a spirit of respect for others' work. The interaction between the teacher's avatar and the learners' avatars played a role in instilling flexibility in students, which influenced students' behavior in the Edu-Metaverse environment, which agrees with the findings in the literature [9,11,16,27,28,38,40,42]. Immersive virtual environments enable interactive assessments that build skills and knowledge.

The second hypothesis tested the differences in the tolerance and respect experienced by the two groups. The results can be attributed to the fact that avatar evaluations heavily

rely on users' preferences and needs. Competitiveness among learner avatars and differences in their perspectives on environment design, as well as the virtual interaction of learner avatars, which was not naturally observed but rather a virtual representation of their personalities, could have affected tolerance levels. As mentioned by Bosch and Ellis [40], tailoring tolerance education to local values and culture can improve social interaction.

Students' avatars may have been evaluated based on the virtual environment rather than on actual comparative avatar levels. The nature of the open and unrestricted virtual metaverse, with freedom in actions without constraints, may have had an impact on the (many-to-one) peer group, increasing the likelihood of students not fully engaging in learning and being prone to distraction and lack of focus. This could explain the emergence of results indicating a lower level of tolerance for avatars in the (many-to-one) group before and after the intervention. The results suggest a decrease in students' tolerance levels after using the metaverse, which differs from the findings in the literature [47–50]. This may be attributed to the difference in the environment: the studies in the literature were carried out in traditional learning environments, such as schools, while the current research study was conducted in the unrestricted virtual environment of the metaverse.

Despite the differences between the two groups in peer evaluation modes and levels of tolerance, the collaboration of peers in using metaverse tools had a positive impact on the design of Edu-Metaverses, and both groups presented environments that explain the concept of sustainability. This aligns with the interpretation of [44,48] that technology can enhance tolerance education by fostering student interaction and empathy, leading to improved educational outcomes. Thanks to the visual, auditory, and interactive features of metaverse technology, students can explore sustainable concepts in innovative and experiential ways, which stimulates curiosity and promotes active learning.

In conclusion, due to the dearth of research on experimental methodologies within metaverse environments, this study proposes innovative assessment approaches congruent with the guidelines of [13,15]. These evaluations aim to assess student performance adeptly in the distinctive realm of the metaverse, all the while fostering values of tolerance and empathy. This aids in elucidating the principles of environmental sustainability within an experimental setting that conserves the natural environment and its resources.

## 5. Conclusions

This study explores the influence of avatars on (one-to-many/many-to-one) groups as an alternative assessment method in the metaverse, on developing a sustainable Edu-Metaverse environment, and on promoting tolerance and respect among graduate female students. The study emphasized that the avatars within the one-to-many group demonstrated a better performance compared to their counterparts (many-to-one) as an alternative mode of assessment in the development of a sustainable Edu-Metaverse environment using a product quality card. Also, the one-to-many group showed an increase in tolerance values after the intervention.

This study may be used as a guide for teachers using the metaverse in their teaching, helping them select the most suitable alternative assessment method when designing an Edu-Metaverse that provides a high degree of creative freedom for learners, without the constraints of time and place, unlike other teaching modes. Using avatars in the Edu-Metaverse as an assessment tool enhances student engagement and focuses on content. The use of avatars in the Edu-Metaverse enables students to express themselves and be evaluated based on their content rather than external factors such as appearance, gender, or race. Hence, this model can promote diversity, reduce biases, and foster tolerance among learners. The study also sheds light on the challenges of integrating sustainable education into the metaverse, which can be beneficial in promoting sustainable practices and shaping a more conscious digital community.

We recommend conducting further research on the impact of the use of avatars of different participants on sustainable Edu-Metaverse platforms, in terms of gender, on educational assessment. Furthermore, conducting studies on different metaverse platforms

across different educational stages could shed light on their impact on learning outcomes and student behaviors, such as quality of life, educational flexibility, and digital well-being. Another suggestion is to conduct a wide range of studies examining the effects of different ethical values in Edu-Metaverses, such as justice, privacy, and ethical artificial intelligence.

**Author Contributions:** Writing—review & editing, Y.A.M.Z.; Funding acquisition, A.A.G. All authors have read and agreed to the published version of the manuscript.

**Funding:** This work was supported by the Deanship of Scientific Research, Vice Presidency for Graduate Studies and Scientific Research, King Faisal University, Saudi Arabia (Project No. KFU241463).

**Institutional Review Board Statement:** This study was conducted by the Declaration of Helsinki and approved by the Research Ethics Committee at King Faisal University (KFU-REC-2024-JUN-ETHICS1829).

**Informed Consent Statement:** Informed consent was obtained from all subjects involved in the study.

**Data Availability Statement:** The original contributions presented in the study are included in the article, further inquiries can be directed to the corresponding author.

**Conflicts of Interest:** The authors declare no conflicts of interest.

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
