# Peer review of "Exploring the Use of Avatars in the Sustainable Edu-Metaverse for an Alternative Assessment: Impact on Tolerance"

_sustainability, doi:10.3390/su16156604_

Round 1

Reviewer 1 Report

Comments and Suggestions for Authors

Though Introduction is well written, my suggestion would be authors to include also a Literature Review section in which they analyse important concepts of their research ( ie metaverse and its significance for education) and in the Introduction section , include important but introductory for their research information and concepts. The Materials and Methods section it is important to include research design information and tools and not theoretical information about the metaverse and avatars. In text citations it is important authors to include the names of the authors’paper -book they are citing and use in the parenthesis the number that the paper has in the paper’s reference list. Again the Materials and Methods section is too long and theoretically oriented- this could be resolved by including a robust literature review section and leave methodology and research design for Materials and Methods section. The research scheme is not clear so authors could describe it in a more clear and systematic manner. Authors refer to specific studies which is good- however it is not clear how do these relate with their research. My suggestion would be to filter respective content and in a more critical manner explore and present how these studies “connect”with their actual work. The methodology selected (quasi experimental)  needs further justification so as to show the added value of it in the specific research. The Discussion section needs to be more extended so as to have a good balance between theoretical information  sections (Introductions and Methods and Materials) and the discussion of data findings in the light of existing literature. The Conclusions section is too long- it could be shorter. Overall this is a very interesting paper discussing about state of the art technology, however its structure and content needs to be filtered so as enhance it in quality and readability flow.

Comments on the Quality of English Language

Academic genre has been applied- no language issues detected. 

Author Response

Comments 1: Though Introduction is well written, my suggestion would be authors to include also a Literature Review section in which they analyze important concepts of their research ( ie metaverse and its significance for education) and in the Introduction section, include important but introductory for their research information and concepts.

Response 1: Done. We agree with this comment.

Comments 2: The Materials and Methods section it is important to include research design information and tools and not theoretical information about the metaverse and avatars.

Response 2: Done.

Comments 3: In text citations it is important authors to include the names of the authors’paper -book they are citing and use in the parenthesis the number that the paper has in the paper’s reference list. Response 3: Done. Thank you for pointing this out.

Comments 4: the Materials and Methods section is too long and theoretically oriented- this could be resolved by including a robust literature review section and leave methodology and research design for Materials and Methods section. 

Response 4: Done.

Comments 5: Authors refer to specific studies which is good- however it is not clear how do these relate with their research. My suggestion would be to filter respective content and in a more critical manner explore and present how these studies “connect” with their actual work.

Response 5: Done.

Comments 6: The methodology selected (quasi-experimental) needs further justification so as to show the added value of it in the specific research.

Response 6: Done. we explain what we do.

Comments 7: The Discussion section needs to be more extended so as to have a good balance between theoretical information sections (Introductions and Methods and Materials) and the discussion of data findings in light of existing literature.

Response 7: Done. accordingly, We modified it to emphasize this point

Comments 8: The Conclusions section is too long- it could be shorter. 

Response 8: Done.

Reviewer 2 Report

Comments and Suggestions for Authors

The manuscript under review, titled "Exploring the Use of Avatar in Sustainable Edu-Metaverse for Alternative Assessment: Impact on Tolerance," demonstrates significant improvements in clarity, methodological rigor, and overall relevance. The abstract effectively summarizes the study by succinctly outlining the methodology, findings, and implications, giving the reader a clear overview of what to expect. The introduction lays a solid foundation by discussing the role of the metaverse in education and the importance of alternative assessment methods. The rationale for the study is well articulated and the objectives clearly stated.

The methodology section has been thoroughly revised, detailing the quasi-experimental design, participant selection and data collection instruments, which increases the transparency and replicability of the study. The results are presented logically with clear tables and figures that facilitate understanding. The statistical analyzes are appropriately performed and well explained, with the discussion section linking the findings to the research questions and hypotheses in an insightful way.

The discussion contextualizes the results within the wider literature and explores the impact of the use of avatars in educational assessments and their potential to promote tolerance and respect among students. The limitations of the study are acknowledged and suggestions for future research are made. The conclusion effectively summarizes the key findings and their implications and reaffirms the study's contributions to the field of educational technology and sustainability.

The bibliography is comprehensive and includes current and relevant literature with accurate citations that meet scholarly standards. Based on these thorough revisions, I recommend this work for publication. The authors have addressed previous concerns and have significantly improved the clarity, methodological rigor, and relevance of the manuscript. The study provides valuable insights into the use of avatars in the edu-metaverse for alternative assessments and their impact on promoting tolerance and respect among students.

Final comments for the authors include ensuring that any minor typos are corrected in the final version. For example, in the abstract, "36 female graduates" should be corrected to "36 female graduates"," and in the introduction, "the students do not have to travel to the schools and thus consume fossil fuels" should be changed to "the students do not have to travel to the schools and thus consume less fossil fuels" Also check that all figures and tables are correctly formatted and referenced in the text. If you make these minor corrections and maintain consistency throughout the manuscript, it will meet the high standards required for publication.

Comments on the Quality of English Language

The quality of the English language in the manuscript is generally good. The text is clear and comprehensible, with well-constructed sentences and appropriate use of academic vocabulary. However, there are some minor typos and places where the wording could be slightly refined for better clarity and flow.

For example:

-In the abstract, "36 female graduate students" should be corrected to "36 female graduate students".

-In the introduction, "the students do not have to travel to schools and therefore consume fossil fuels" could be changed to "the students do not have to travel to schools and therefore consume less fossil fuels".

Author Response

Point 1: Final comments for the authors include ensuring that any minor typos are corrected in the final version. For example, in the abstract, "36 female graduates" should be corrected to "36 female graduates"," and in the introduction, "the students do not have to travel to the schools and thus consume fossil fuels" should be changed to " the students do not have to travel to the schools and thus consume less fossil fuels"

Response 1: Done. Thank you for pointing this out.

Point 2: Also check that all figures and tables are correctly formatted and referenced in the text.

Response 2: Done. Thank you for pointing this out.

Round 2

Reviewer 1 Report

Comments and Suggestions for Authors

In the Abstract section some information on the preliminary findings and aspects of research design would be helpful.The concept of tolerance which is quite important for the paper, could be presented and analysed in earlier paragraphs of the Introduction section, providing some kind of introduction to the reader. Better filtering of content could lead to the use of Metaverse related content and Avatar related content to specific sections of the paper so as to increase cohesion and coherence of the paper. However, the Literature Review section further enhances the quality of the content presented. A justification of quasi experimental methodology could be included. The Discussion section in terms of content presentation could be better aligned with respective literature in the Literature Review section. The Conclusion section could be shorter.

Author Response

Response to Reviewer 1 Comments

We appreciate your precious time in reviewing our paper and providing valuable comments, Thanks !

Comments 1: In the Abstract section some information on the preliminary findings and aspects of research design would be helpful.

Response 1: Done. We agree with this comment.

Comments 2: The concept of tolerance which is quite important for the paper, could be presented and analysed in earlier paragraphs of the Introduction section, providing some kind of introduction to the reader.

Response 2: Done.

Comments 3: Better filtering of content could lead to the use of Metaverse related content and Avatar related content to specific sections of the paper so as to increase cohesion and coherence of the paper. However, the Literature Review section further enhances the quality of the content presented.

Response 3: Done.

Comments 4: A justification of quasi experimental methodology could be included.

Response 4: To investigate causal relationships within the metaverse, a virtual environment mirroring the real world, A quasi-experimental design was employed to evaluate performance. Two groups of postgraduate students enrolled in a "Three-Dimensional Educational Graphics" course were compared. Participants had no background in using metaverse, ranged in age from 22 to 35 years, and were assigned to groups based on existing class sections. To establish group equivalence, a pre-test using the IPTS measure was administered (Table 1).

Badiee, M., Wang, S. C., & Creswell, J. W. (2012). Designing community-based mixed methods research. In D. K. Nagata, L. Kohn-Wood, & L. A. Suzuki (Eds.), Qualitative strategies for ethnocultural research (pp. 41–59). American Psychological Association. https://doi.org/10.1037/13742-003

KUÅš, A. (2024). USING QUASI-EXPERIMENTAL DESIGNS FOR CAUSAL EFFECTS. Scientific Papers of Silesian University of Technology. Organization & Management/Zeszyty Naukowe Politechniki Slaskiej. Seria Organizacji i Zarzadzanie, (194).‏ https://doi.org/10.29119/1641-3466.2024.194.12 .

Comments 5: The Discussion section in terms of content presentation could be better aligned with respective literature in the Literature Review section.

Response 5: Done.

Comments 6: The Conclusion section could be shorter.

Response 6: Done.
